# All Politics is Local: Redistricting via Local Fairness

**Shao-Heng Ko**[*]     **Erin Taylor**[*]     **Pankaj K. Agarwal**     **Kamesh Munagala**

Department of Computer Science
Duke University
Durham, North Carolina 27708 USA
Email: {sk684,ect15,pankaj,kamesh}@cs.duke.edu

## Abstract

In this paper, we propose to use the concept of *local fairness* for auditing and ranking redistricting plans. Given a redistricting plan, a *deviating group* is a population-balanced contiguous region in which a majority of individuals are of the same interest and in the minority of their respective districts; such a set of individuals have a justified complaint with how the redistricting plan was drawn. A redistricting plan with no deviating groups is called *locally fair*. We show that the problem of auditing a given plan for local fairness is NP-complete. We present an MCMC approach for auditing as well as ranking redistricting plans. We also present a dynamic programming based algorithm for the auditing problem that we use to demonstrate the efficacy of our MCMC approach. Using these tools, we test local fairness on real-world election data, showing that it is indeed possible to find plans that are almost or exactly locally fair. Further, we show that such plans can be generated while sacrificing very little in terms of compactness and existing fairness measures such as competitiveness of the districts or seat shares of the plans.

## 1   Introduction

Redistricting in the United States is the process of partitioning a state into districts, each of which elects one representative to the Congress, for the most part, via simple majority voting. As of April 2022, one year after the US Census Bureau released the results of the 2020 decennial census, 41 out of the 50 states have finished redrawing the congressional redistricting plans for the next decade [29]. This process has triggered numerous debates and litigation along the way. Much of this debate centers on whether the plans are *gerrymandered* so that one of the two parties gets more representatives. Given its high-stakes impact and mathematical richness, there has been persistent interest in tackling redistricting as an algorithmic question since the early 1960s [7].

There is ongoing debate around what a "desirable" redistricting plan should be. It is commonly agreed that "desirable" plans should, at minimum, produce population balanced, contiguous, and compact districts [1]. Beyond this basic agreement, there is still debate on richer notions of desirability, particularly notions related to the "fairness" of a plan. This has motivated a long line of recent work [14, 15, 23] as well as software tools [4, 29] on *auditing* a given redistricting plan against various fairness concepts. Some of these concepts have since been adopted in Wisconsin's and Michigan's redistricting efforts [11]. It should be noted that under most notions of desirability proposed in literature, the problem of redistricting is computationally hard [27], leading to the study of heuristic approaches, which we outline later.

**Global versus Local Fairness.** Zooming into fairness criteria, most extant notions of fairness focus on the *global outcomes* of the redistricting plans, e.g., whether the *seat shares* proportionally represent the demographics [41], or how competitive the districts drawn are [14]. However, it is argued in [5]

---

[*]Equal contributions.

36th Conference on Neural Information Processing Systems (NeurIPS 2022).

that global metrics do not always distinguish between *natural* gerrymandering – when the distribution of voters unavoidably prohibits certain globally fair outcomes – and *artificial* gerrymandering – when the plans are manipulated to favor a demographic group. This issue is typically addressed via *statistical tests* [15]: a probabilistic method is used to generate an ensemble of population balanced, contiguous, and compact plans, and the global fairness score in question is computed for each of these plans, yielding a histogram of scores. The plan in question is deemed "fair" if its global fairness score is not an outlier in this histogram.

Global fairness, such as proportional seat shares, despite being desirable and statistically testable, may not represent the *local* concerns of voters. For instance, imagine the blue party cares about rising sea levels and climate change, while the red party does not. In North Carolina, if at least one seat on the eastern coast has blue majority, that representative may advocate to mitigate the impacts of climate change to the coastal residents on the state or federal level. On the other hand, a better seat share may lead to a plan in which all districts near the coast have red majority, while the districts in the western mountains have blue majority. However, the latter set of representatives may not advocate for issues impacting the coastal residents, since it is not of local concern to the geographic area. This motivates the need for local fairness as a separate fairness measure, capturing at some level the saying "all politics is local".

Borrowing the notion of core from cooperative game theory, the work of [5] defines local fairness notion as follows: given a redistricting plan, a voter is unsatisfied if the majority demographic in her district does not match her own demographic. A redistricting plan is locally fair if no group of unsatisfied voters could *deviate* and draw a different district such that this group of unsatisfied voters has a majority in the new district.

As in the scenario above, such a local notion of fairness has the advantage of capturing *justified complaints* of groups of voters, as has happened in earlier court judgements [2]. It also provides a way of auditing enacted plans without resorting to statistical tests, making it more human interpretable and *explainable*.

**Research Questions.** The notion of local fairness is appealing; however, the analysis and results in [5] are theoretical and apply only to a simplified one-dimensional model. In this paper, we develop algorithms to audit plans for local fairness, and systematically study this concept on real-world electoral data. In particular, we study the following questions:

- Given a redistricting plan, can we efficiently test (or audit) whether the plan is locally fair?
- Are locally fair plans achievable in real redistricting tasks? If not, can we quantify how far a given plan is from being locally fair?
- Is local fairness empirically compatible with other existing global fairness concepts?

## 1.1 Related Work

**Redistricting as Optimization.** We first focus on the task of drawing plans, or computational redistricting. The idea of using computational tools in redistricting dates back to the 1960s [24, 39]. Since then, an extensive line of work (see [7] for a comprehensive survey) cast the redistricting task as an optimization problem, in which the input contains only spatial location of individuals, but not their political affiliations. The objective and constraints capture the population balance, contiguity, and compactness criteria of the districts. This problem is computationally intractable in the worst case [13], and multiple algorithmic approaches have been proposed, including Voronoi diagrams [20, 28], local search [26], simulated-annealing and hill climbing [6], and spatial evolutionary algorithms [31]. On the flip side, it is argued in [10, 42] that such "neutral" districting plans – as outputs of algorithms without political inputs – may contain unintentional biases, as well as unexpected outcomes such as "natural gerrymandering" [8, 19], i.e., the geographic distributions of voters naturally lead to disproportionate seat shares. Therefore, fairness objectives such as partisan representativeness are typically incorporated into the redistricting problem as objectives; however, these additional requirements further add to the computational difficulty of the problem [27].

**Ensemble Approaches to Redistricting.** Instead of optimizing and finding a single best redistricting plan, another line of work focuses on generating a large *ensemble* of districting plans, with the hope of some of these plans being fair. These methods include Flood Fill [12, 32], Column Generation [21], and the widely adopted Markov Chain Monte Carlo (MCMC) approach [18, 30, 38]. The latter

approach *samples* from the space of feasible plans with a bias towards "desirable" or fairness properties. For instance, it is shown in [35] that the widely used ReCom MCMC method [15] provably biases towards compact plans. We provide a more in-depth description of ReCom in Section 3. The work of [17] proposes a method for choosing one representative plan from such an ensemble based on defining distances between plans.

**Auditing and Combating Gerrymandering.** A somewhat different question from constructing a desirable plan is the question of *auditing* a given plan for desirability and fairness. As mentioned before, ensemble based approaches provide a natural, statistical way of auditing [22, 23]: The properties of the enacted plan is compared against the histogram of the corresponding property on the ensemble; if the plan is a statistical outlier, then it is considered more "gerrymandered" and hence less desirable. The recent work of [30] instead uses plans in the ensemble as comparators to identify manipulation in redistricting plans. On the non-statistical side, numerous approaches to auditing have also been proposed via appropriate desirability scores. These are either scores based on compactness of the plan (such as the Reock [36] and Polsby-Popper [34] scores), or scores based on partisan outcomes generated by the plan (such as the efficiency gap [37], mean-median gap [40], partisan symmetry [41], and the GEO metric [9]), or scores based on competitiveness of the plan [14]. Many of these measures are used in publicly available tools [3, 4]. Finally, there is a recent line of work that attempts to eliminate gerrymandering by completely revamping the winner-takes-all, single-member district mechanism into a multiwinner election [19].

## 1.2 Our Contribution

In this paper, we take the standard view of redistricting as partitioning a planar graph on precincts into population-balanced, contiguous, and (in a heuristic sense) compact regions. We naturally extend the local fairness concept proposed in [5] to this task.

We first focus on the question of *auditing* a given plan for local fairness, that is, the non-existence of a population-balanced contiguous region in which a majority of voters are of the same party and is minority in the given plan. We show that this problem is computationally intractable in the worst case. Our first contribution is two heuristics for the auditing problem. Our first approach, that is scalable and practical, extends existing ensemble-based methods in a novel way: we assume the districts in the ensemble are the only districts to which voters can deviate, and given a plan to be audited, we test each of these districts as a potential deviation on that plan. Our second approach drills deeper into plans where the ensemble based method finds no deviating group; indeed, if the method found a deviating group, the plan was already deemed not locally fair. On the former set, we generate several random spanning trees, and devise a polynomial time dynamic programming algorithm that audits each tree for local fairness. If any of these audits finds a deviating group, the original plan was not locally fair. The dynamic program is not as efficient as the ensemble-based method; however, we provide empirical evidence that the ensemble method suffices to deem a plan locally fair, and the dynamic program typically does not find additional compact deviating groups. Finally, for redistricting plans that are not locally fair, we propose a measure that quantifies the unfairness of the plans by the portion of population with a justified complaint.

As our second contribution, we empirically study the notion of local fairness on real data on recent elections in the US. We generate plans using the (by now) standard ReCom [15] ensemble method, and audit each plan for local fairness using the ensemble method, thereby producing an ordering of the plans via our unfairness measure. We empirically show that applying the criterion of local fairness prunes the space of candidate plans considerably, while still returning a set of potential candidates. Most global and statistical notions of fairness fail to do such pruning, since they are endogenously defined relative to the order statistics on the ensemble. We further show that not only is local fairness *achievable* on real redistricting tasks, but it is also compatible with extant global fairness properties. Indeed, when we compare locally fair plans and those with many deviating groups, the former tend to be just as compact, have comparable seat share outcomes, and sacrifice only a small amount of competitiveness. Thus local fairness can be used as an additional fairness criterion in conjunction with a global fairness criterion. We also investigate robustness of the local fairness concept, and show that fair redistricting plans remain consistent across different elections used. We finally show visualizations of fair and unfair plans; in particular showing that the visualization of deviating groups makes the local fairness notion explainable.

Taken together, our results demonstrate local fairness as an effective *pruning criterion* for candidate redistricting plans while sacrificing little in other desired properties. We also note that in practice, there could be other considerations when choosing the "best" plan even among many locally-fair plans; we leave the question of choosing these considerations to policy makers.

## 2 Model and Preliminaries

In keeping with recent literature [13, 15, 16, 21], the input to the redistricting problem is a planar connected graph $G = (V, E)$ where each vertex $v \in V$ represents an indivisible geographic unit (a precinct or a census block),[2] and an edge is placed between two vertices if they are geographically adjacent. Going forward, we refer to each $v \in V$ as a *precinct* and $G$ as the *precinct graph*.

**Redistricting Plans.** For each precinct $v \in V$, let $\rho(v) > 0$ denote its population and let $\tau(v) \in [0, \rho(v)]$ denote the number of voters in $v$.[3] We let $\gamma(v) \in [0, 1]$ and $\beta(v) = 1 - \gamma(v)$ denote the fraction of $\tau(v)$ who vote *red* and *blue*, respectively. Note that it is assumed each individual voter is exactly one of the two colors. For an arbitrary subset of precincts $W \subseteq V$, set $\rho(W) := \sum_{v \in W} \rho(v)$, $\tau(W) := \sum_{v \in W} \tau(v)$, $\gamma(W) := \frac{\sum_{v \in W}(\gamma(v) \cdot \tau(v))}{\tau(W)}$, and $\beta(W) := 1 - \gamma(W)$.

**Definition 1** (*k*-redistricting plan). *A $k$-redistricting plan of $G = (V, E)$ is a partition of $V$ into $k$ pairwise-disjoint subsets $D_1, D_2, \ldots, D_k \subseteq V$, called districts. Each district assumes the color of the majority of its voters. For a redistricting plan $\Pi$, let $B_\Pi$ (resp. $R_\Pi$) denote the set of precincts in blue (resp. red) districts in $\Pi$.*

In the following, we fix an error parameter $\varepsilon > 0$, and the desired number of partitions $k$. Note that the average population per district is $\frac{\rho(V)}{k}$. We say a district $D \subseteq V$ is $\varepsilon$-*feasible* if: (1) $D$ induces a connected subgraph, and (2) the population of $D$ is at most $\varepsilon$ away from average, *i.e.*, $(1 - \varepsilon) \cdot \frac{\rho(V)}{k} \leq \rho(D) \leq (1 + \varepsilon) \cdot \frac{\rho(V)}{k}$. A redistricting plan $\Pi$ is $\varepsilon$-*feasible* if each district $D_i \in \Pi$ is $\varepsilon$-feasible.

We note that this definition of an $\varepsilon$-feasible plan is consistent with the general practice in the U.S, where the sizes of districts should be balanced in terms of their population, based on census information, not in terms of the number of eligible voters. Since $\varepsilon$ and $k$ will be fixed throughout, we drop the prefixes and refer to $k$-redistritcting plans and $\varepsilon$-feasible districts as *redistricting plans* and *feasible* districts, respectively.

**Local Fairness.** We extend the notion of *local fairness* proposed by [5] to the graph-based redistricting problem. We say that a feasible district $W \subseteq V$ is *red-majority* (*red* in short) if $\gamma(W) \geq \beta(W)$, and *blue-majority* (*blue*) otherwise. We call this majority color as the *color* of $W$. Given a redistricting plan $\Pi$, any voter whose color agrees with the color of its assigned district in $\Pi$ is deemed *happy* with respect to $\Pi$, and the remaining voters are *unhappy*.

**Definition 2** (*c*-locally fair). *Given a feasible redistricting plan $\Pi$ of $G$ and a constant $c \in [1/2, 1]$, a feasible district $W \subseteq V$ is a red $c$-deviating group with respect to $\Pi$ if $W$ is red and at least a $c$-fraction of its voters are unhappy red voters in $\Pi$, or formally, $\sum_{v \in W \cap B_\Pi} \gamma(v) \cdot \tau(v) > c \cdot \tau(W)$. A blue $c$-deviating group is defined analogously. We call a feasible redistricting plan $\Pi$ of $G$ $c$-locally fair if there are no red or blue $c$-deviating groups with respect to $\Pi$.*

When $c = 1/2$, only a simple majority of voters in a deviating group must be unhappy. In this special case, we omit the prefix $c$ by referring to red deviating groups, blue deviating groups, and locally fair redistricting plans. Throughout, we refer to locally fair redistricting plans as fair plans.

We are thus interested in the following two problems.

**LF AUDITING problem.** Given a feasible redistricting plan $\Pi$ and a parameter $c \in [1/2, 1]$, decide whether $\Pi$ is $c$-locally fair.

**LFP GENERATION problem.** Given a precinct graph $G$ and parameters $\varepsilon, k$ and $c$, compute a feasible redistricting plan $\Pi$ of $G$ such that $\Pi$ is $c$-locally fair, or report that none exists.

---

[2]Typically, precincts are not split by redistricting plans [15].

[3]We assume that we know the exact number of people who cast a vote in each precinct, along with which candidate they voted for, such as is available for historical elections

For a redistricting plan $\Pi$ that is not locally fair, we quantify its degree of unfairness as follows. Consider all deviating groups of $\Pi$, and define the unfairness score of $\Pi$ as the fraction of all voters that are unhappy in some deviating group. Formally, let

$$W_{\text{B}}^*(\Pi) := R_\Pi \cap \left( \bigcup \left\{ W \subseteq V \mid W \text{ is a blue deviating group of } \Pi \right\} \right)$$

denote the set of red precincts that lie in some blue deviating group of $\Pi$. Similarly, define $W_{\text{R}}^*(\Pi)$. Then the unfairness score of $\Pi$ is defined as:

$$\text{unf}(\Pi) := \frac{\beta(W_{\text{B}}^*(\Pi)) \cdot \tau(W_{\text{B}}^*(\Pi)) + \gamma(W_{\text{R}}^*(\Pi)) \cdot \tau(W_{\text{R}}^*(\Pi))}{\tau(V)}.$$

This score captures the fraction of voters that are both (i) unhappy in $\Pi$ and (ii) in the majority color of some deviating group of $\Pi$. Note that $\text{unf}(\Pi) \in [0, 1]$, and equals zero if $\Pi$ is locally fair.

**Compactness.** In addition to requiring that districts be contiguous and population balanced, many redistricting models also require that the districts be *compact*. However, in contrast to the former two criteria, there is no universally agreed measure of compactness [7, 21]. For example, the Princeton Redistricting Report Cards[4] uses Reock [36] and Polsby-Popper [34] scores, both of which are derived from the area and perimeter of the geographic districts drawn. It also uses the number of counties split into multiple districts. In the discrete model of precinct graphs, one common measure of compactness is the number of cut edges formed by the plan, i.e., the number of edges whose endpoints lie in different districts of $\Pi$ [13, 15]. Though we do not enforce compactness in the generation and audit problems, the algorithms we use are biased towards compact plans, as we empirically demonstrate.

**Organization.** In Section 3, we present two algorithms for the LF AUDITING and LFP GENERATION problems. We then describe the experimental setup and empirical results in Section 4. Additional methodological and experimental results are presented in the Supplementary Material.

# 3   Heuristics for Efficiently Auditing Local Fairness

In Appendix A, we prove that both the LF AUDITING and LFP GENERATION problem are NP-complete, thereby necessitating heuristic or approximately optimal approaches.

Our main contribution in this section is efficient heuristics for LF AUDITING. The methods trade off computational efficiency for accuracy in determining local fairness. We describe the types of inaccuracy (one-sided versus two-sided error) that arise after describing the methods. We also provide empirical evidence that the more computationally efficient of these methods suffices to deem plans as locally fair. For LFP GENERATION, we simply generate an ensemble of feasible redistricting plans using existing MCMC approaches, and run the LF AUDITING algorithm to find the unfairness score $\text{unf}(\Pi)$ for each generated plan $\Pi$, thereby ranking the plans in the ensemble in terms of fairness.

## 3.1   Ensemble-based Auditing

The computationally more efficient approach makes use of ensembles in a novel way as follows:

1. **Generation.** Generate an ensemble of $t$ redistricting plans $\mathcal{E} = \{\Pi_1, \Pi_2, \ldots, \Pi_t\}$.
2. **Districts to test.** Let $\Delta = \bigcup_{j=1}^t \Pi_j$ be the collection of districts in the plans in $\mathcal{E}$.
3. **Audit.** Treat $\Delta$ as the candidate set of deviating groups. For each plan $\Pi \in \mathcal{E}$, compute if each $D \in \Delta$ is a $c$-deviating group (either red or blue). This step yields, for each $\Pi \in \mathcal{E}$, the set $\tilde{W}_{\text{B}}^*(\Pi) \subseteq V$ of precincts in blue deviating groups $D \in \Delta$ (which we use as an approximation of $W_{\text{B}}^*(\Pi)$), and similarly $\tilde{W}_{\text{R}}^*(\Pi)$. We then use $\tilde{W}_{\text{B}}^*(\Pi)$ and $\tilde{W}_{\text{R}}^*(\Pi)$ to compute the unfairness score $\text{unf}(\Pi)$ for each $\Pi \in \mathcal{E}$.

For the first step of an ensemble of plans $\mathcal{E}$, we use the ReCom algorithm [15]. This algorithm develops a Markov Chain whose state space is the full set of ($\varepsilon$-)feasible ($k$-)partitions of $G$. In each step, it randomly combines two (or more in its general form) districts of the current redistricting plan,

---

[4] https://gerrymander.princeton.edu/

generates a random spanning tree of the subgraph induced by the combined districts, and re-partitions the subgraph into two parts by cutting edges in the spanning tree so that the new resulting districts remain population balanced. The random spanning tree step *biases* the Markov chain towards plans with compact districts, where compactness is measured by number of cut edges [35].

In the remainder of the paper, we describe our auditing framework when `ReCom` is used as the ensemble generation technique. We note however that we can use any other method that produces compact plans, for instance, iterative-merging flood-fill algorithms [10].

## 3.2 Auditing via Dynamic Programming on Trees

The ensemble-based LF AUDITING method makes one-sided error – if it finds a deviating group, then the plan is not fair, while the absence of a deviating group from $\Delta$ only means the plan is *likely* fair. Indeed, in our experiments, there are often multiple redistricting plans that the ensemble-based audit deems likely fair, calling for a more systematic method for further analyzing the candidate plans that are potentially fair. This motivates the approach we now describe.

We show that the LF AUDITING problem is efficiently solvable if $G$ is a tree. We then exploit this fact and use this as a heuristic to solve LF AUDITING on a general planar graph $G$, given partition $\Pi$, as follows:

1. **Random spanning trees.** First generate a collection of random spanning trees $\mathcal{T}$ of $G$.
2. **Audit each tree.** For each tree $T \in \mathcal{T}$, decide if there exists a feasible deviating group with respect to $\Pi$ such that this district is a connected subtree of $T$. Note that such a district will also be a connected subgraph in $G$.

We implement step 2 for a tree $T$ using dynamic programming. We check for the existence of a blue deviating group for $c = 1/2$ in a tree $T$ as follows; the case for red groups and larger $c$ follows similarly. Note that the range for the size of a district is $[(1-\varepsilon)\sigma, (1+\varepsilon)\sigma]$, where $\sigma = \rho(V)/k$. For each precinct $v \in V$, let $D(v)$ denote the district that contains $v$ in $\Pi$. Let the number of unhappy blue individuals in precinct $v$ be denoted $\mathrm{uhp}(v) = \beta(v) \cdot \tau(v)$ if $D(v) \in R_\Pi$, and zero otherwise. The task is to decide whether there is a connected subtree $W \subset T$ such that $\sum_{v \in W} \rho(v) \in [(1-\varepsilon)\sigma, (1+\varepsilon)\sigma]$ and $\sum_{v \in W} \mathrm{uhp}(v) > \frac{1}{2} \cdot \sum_{v \in W} \tau(v)$.

For each $v \in V$, let $T_v$ be the subtree of $T$ rooted at $v$. For a vertex $v \in V$ and two parameters $i, p \leq (1+\varepsilon)\sigma$, let $A[v, i, p]$ denote the maximum number of unhappy blue voters in a subtree $W \subseteq T_v$ such that $v \in W$, $\tau(W) \leq i$, and $\rho(W) = p$. To compute $A[v, i, p]$, we take the best subset of children of $v$ such that unhappy blue voters are maximized subject to population and voter constraints.

For a leaf $v$ of $T$, $A[v, i, p] = \mathrm{uhp}(v)$ if $i \geq \tau(v)$ and $p = \rho(v)$, and $A[v, i, p] = 0$ otherwise.

Next, let $v$ be an internal node of $T$. Let $U = \mathrm{children}(v) = \{u_1, \ldots, u_{\deg(v)}\}$ denote the set of $v$'s children in $T$. Let $i' = i - \tau(v)$ and $p' = p - \rho(v)$. We have

$$A[v, i, p] = \mathrm{uhp}(v) + \max_{\{i_j, p_j\}} \left\{ \sum_{j=1}^{\deg(v)} A[u_j, i_j, p_j] \, \Big| \, \left( \sum_{j=1}^{\deg(v)} i_j \leq i' \right) \wedge \left( \sum_{j=1}^{\deg(v)} p_j = p' \right) \right\}. \quad (1)$$

To compute the second term in Equation (1) efficiently, let $B_{v,i,p}[j, x, y]$ denote the maximum number of total unhappy blue voters in a union of subtrees rooted at $\{u_1, \ldots, u_j\}$ with total voter count at most $x$ and total population level $y$. The second term of the RHS in Equation (1) is $B_{v,i,p}[\deg(v), i', p']$. We then have $B_{v,i,p}[1, x, y] = A[u_1, x, y]$ for all $x, y$, and for all $j \geq 2$ we have

$$B_{v,i,p}[j, x, y] = \max_{x' \in [0,x], y' \in [0,y]} \left\{ B_{v,i,p}[j-1, x-x', y-y'] + A[u_j, x', y'] \right\}. \quad (2)$$

The algorithm thus proceeds by computing all $A[v, i, p]$ in an outer loop in increasing order of $i$ and $p$ and in bottom-up order of $v$. Each computation of $A[v, i, p]$ proceeds with an inner loop that computes $B_{v,i,p}[j, x, y]$ in increasing order of $j$, $x$, and $y$. For each $v$, after computing all $A[v, i, p]$, we check if there is any $p \in [(1-\varepsilon)\sigma, (1+\varepsilon)\sigma]$ such that $A[v, i, p] > \frac{i}{2}$. If so, there exists a blue deviating group rooted at $v$ with population $p$, total voter count at most $i$, and total number of unhappy blue voters $A[v, i, p]$.

**Time complexity.** The algorithm computes $O(|V| \cdot \sigma^2)$ values of $A[v, i, p]$, each computing $O(\deg(T) \cdot \sigma^2)$ values of $B_{v,i,p}[j, x, y]$. Computing each $B_{v,i,p}[j, x, y]$ requires $O(\sigma^2)$-time to loop through all values of $x'$ and $y'$. Hence the overall time complexity is $O(|V| \cdot \sigma^6 \cdot \deg(T))$. We conclude the following.

**Theorem 1.** LF AUDITING *problem on a tree* $T(V, E)$ *can be solved in time* $O(|V| \cdot \sigma^6 \cdot deg(T))$. *Here,* $deg(T)$ *is the maximum degree of a node in* $T$.

**Ensemble-based Auditing is Approximately Sufficient**   The dynamic programming method is less desirable because of two significant limitations. The first is its running time, which is prohibitively high for practical use. The second limitation is the introduction of a different type of error from the ensemble-based method. Note that as with the ensemble-based method, the non-existence of a deviating group in the DP only provides high confidence (but not absolute certainty) about the plan being locally fair. At the same time, there is a more subtle type of error the DP can make in the other direction: even if it finds a deviating group on the tree, this may not be a *compact* deviating group in the original graph. Therefore, the algorithm can output "not locally fair" when the only deviating groups are non-compact. We therefore need a final step where we check the deviating group to see that it is both feasible and compact on the original graph.

In Appendix B.1, we first show approaches to speed up the running time of the dynamic program to make it practical. In Appendix B.2, we describe additional assumptions under which the dynamic program runs efficiently, we empirically show that it fails to find compact $c'$-deviating groups in plans that the ensemble-based audit method deems $c$-locally fair, for $c'$ slightly larger than $c$. On the other hand, it easily finds such deviating groups in plans that the ensemble-based method deems unfair. In summary, the dynamic programming approach obviates its own use in practice, since we provide strong evidence that the ensemble-based auditing method suffices in order to deem redistricting plans as locally fair, assuming the strength $c$ of the deviating group is relaxed slightly.

## 4   Experiments

In our experiments, we attempt to answer the following questions. First, is local fairness achievable on real redistricting tasks? Second, is it compatible with extant measures of global fairness? Finally, is the notion robust if the underlying data changes? Given the previous discussion, our experiments in this section focus exclusively on the ensemble based method. In Appendix B we discuss the experiments for auditing via dynamic programming.

### 4.1   Datasets and Methods

All data used in our experiments is obtained from the MGGG States open repository [33]. We obtain shapefiles and precinct graphs for Massachusetts (MA), Maryland (MD), Michigan (MI), North Carolina (NC), Pennsylvania (PA), Texas (TX), and Wisconsin (WI).[5] We set $\rho(v)$ to the 2010 census total population in each precinct $v$. The default election we use is the 2016 presidential election, while the 2012 presidential election is also used in the robustness tests. For each precinct $v$, we collect the number of total votes for the Republican party $r_v$ and the total vote amount for the Democrat party $b_v$ in the 2016 presidential election from [33]. We set $\gamma(v) = r_v/(r_v + b_v)$, $\beta(v) = 1 - r_v$, and $\tau(v) = r_v + b_v$, so $\tau(v)$ is the total number of red (Republican) and blue (Democrat) voters.

Using the ReCom algorithm, we generate an ensemble of redistricting plans for each state.[6] Each ensemble consists of $1,000$ redistricting plans, each being the outcome of an independent $10,000$-step Markov chain (with default population balance parameter $\varepsilon = 0.02$) seeded with a recent congressional electoral plan of the state. We set $k$ to be the number of congressional districts in the 2016 election in each state. We then obtain the collection $\Delta$ of candidate districts for each state by taking the union of the districts in each plan in the ensemble. The properties of input graphs of states and their corresponding ensembles are summarized in Table 1.

| State | $|V|$ | $|E|$ | $\rho(V)$ | $\tau(V)$ | $k$ | $|\Delta|$ |
|---|---|---|---|---|---|---|
| MA | 2.1K | 5.9K | 6.55M | 5.13M | 9 | 8.5K |
| MD | 1.8K | 4.7K | 5.77M | 4.42M | 8 | 7.8K |
| MI | 4.8K | 12.5K | 9.88M | 7.54M | 14 | 14.0K |
| NC | 2.7K | 7.6K | 9.53M | 7.25M | 13 | 12.9K |
| TX | 8.9K | 24.7K | 25.14M | 18.28M | 36 | 35.9K |
| PA | 9.2K | 25.7K | 12.70M | 9.91M | 18 | 18.0K |
| WI | 7.1K | 19.5K | 5.69M | 4.35M | 8 | 7.87K |

Table 1: Properties of data and ensembles.

| State | MA | MD | MI | NC | PA | TX | WI |
|---|---|---|---|---|---|---|---|
| $c = .5$ | 100% | 26.7% | 0.3% | 4.9% | 12.0% | 2.8% | 76.9% |
| $c = .51$ | 100% | 27.0% | 1.6% | 8.7% | 28.6% | 5.4% | 83.6% |
| $c = .52$ | 100% | 28.0% | 11.9% | 14.8% | 41.4% | 8.7% | 87.2% |
| $c = .55$ | 100% | 31.5% | 35.2% | 63.9% | 95.2% | 26.0% | 94.0% |

Table 2: Percent of plans without $c$-deviating groups for different $c$ values.

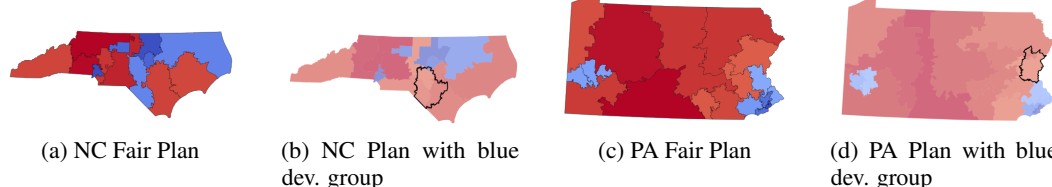

(a) NC Fair Plan

(b) NC Plan with blue dev. group

(c) PA Fair Plan

(d) PA Plan with blue dev. group

Figure 1: North Carolina and Pennsylvania plans without and with deviating groups. The blue deviating groups are drawn with a black outline. In these figures, the districts are coded by its color and the extent of partisanship: districts with a larger value of $\gamma$ (resp. $\beta$) are colored in darker red (resp. blue).

## 4.2 Locally Fair Plans: Counts and Visualizations

We first use the ensemble-based audit method to audit each plan in the ensemble against the set $\Delta$ of districts from the entire ensemble. For each $c \in \{0.5, 0.51, 0.52, 0.55\}$, we count the number of plans in the ensemble without $c$-deviating groups in $\Delta$. We present the results in Table 2. For all values of $c$, there exists plans in the ensemble that admit no $c$-deviating groups in $\Delta$ and thus are identified locally fair by the ensemble-based algorithm. Clearly, a larger value of $c$ implies more plans are identified as locally fair. With $c = 1/2$, very few (but a non-zero number of) plans are identified as fair in four of the seven states. Every plan in the MA ensemble is identified as fair with all districts won by one party. Hence, we omit MA from all subsequent experiments.

In Figures 1a and 1c, we present examples of fair plans (with no $0.5$-deviating groups) in NC and PA respectively, while in Figures 1b and 1d, we present "unfair" plans with many $0.5$-deviating groups. We show visualizations for other states in Appendix E.

We note that a large fraction of plans are locally fair in some states (WI), while others have many deviating groups, even for a large setting of $c$ (MI). To understand where deviating groups are located, in Figure 2, we plot a heat-map of the precincts, counting the number of $1/2$-deviating groups (of either color) that contain that precinct, with yellow representing large counts and purple representing low counts. In every state except MD,[7] we observe that precincts with highest counts are located either in an urban area or in proximity of one. This phenomenon is consistent with the perceived correlation between voter distribution and type of residence [8, 42]. In states with multiple dense urban areas (e.g., NC), there is sufficient flexibility in the redistricting process to "crack" a highly-concentrated

---

[5]These are chosen to represent a spectrum from states whose elections are typically competitive (e.g., NC and WI) to states whose elections are typically lopsided (e.g., MA and TX).

[6]Note that ReCom generates plans without taking into account electoral data.

[7]We discuss MD in more detail in Appendix E.

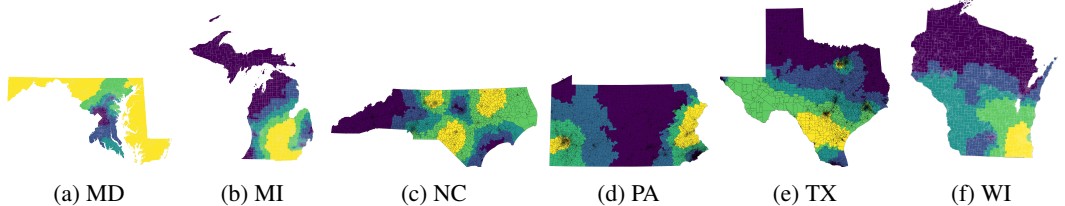

| (a) MD | (b) MI | (c) NC | (d) PA | (e) TX | (f) WI |

Figure 2: Precincts shaded in lighter color are contained in more deviating groups.

urban demographic into multiple districts. In this case, the urban area may form a deviating group resulting in high numbers of unfair plans.

We note that our visualizations – in particular, the deviating groups in unfair plans, as well as the heat-map of likelihood of unhappiness of a precinct – make the local fairness concept *explainable*.

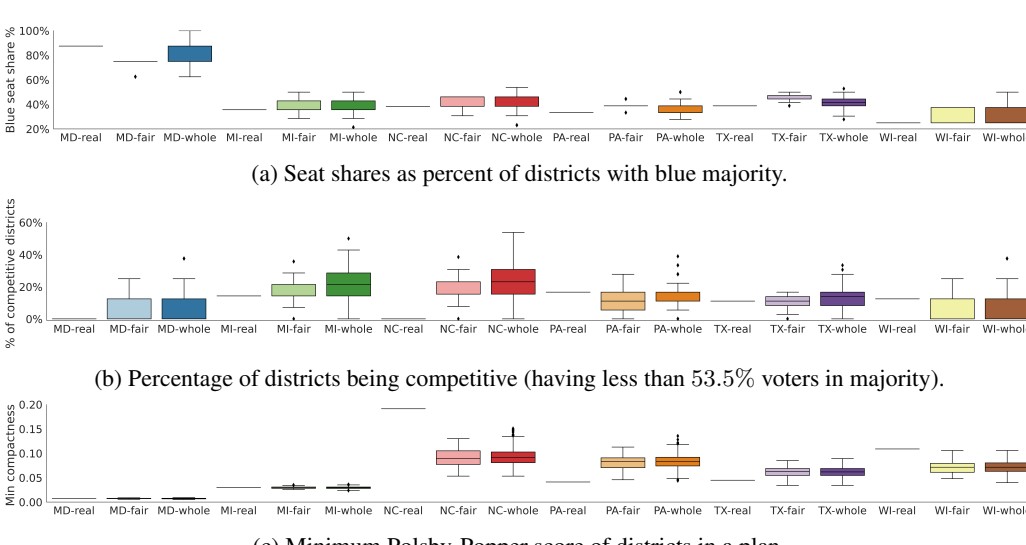

(a) Seat shares as percent of districts with blue majority.

(b) Percentage of districts being competitive (having less than $53.5\%$ voters in majority).

(c) Minimum Polsby-Popper score of districts in a plan.

Figure 3: Distribution of fairness and compactness metrics among subsets of generated plans

### 4.3 Compatibility with Extant Fairness and Compactness Notions

The ensemble-based auditing approach can be viewed as a pruning method that identifies a subset of plans as locally fair. We now ask: how do locally fair plans perform on extant global fairness and compactness criteria compared to average plans in the ensemble? Towards this end, for $c = 1/2$, we rank the plans in the ensemble by the unfairness score $\text{unf}(\Pi)$. We compare properties of the top $5\%$ plans in the ranking (which are most locally fair) against the entire ensemble, as well as a recent enacted congressional redistricting plan.[8]

**Seat share outcomes.** For each plan, we compute *Blue%*, the percentage of seat shares claimed by the Democratic candidate. The resulting distribution is shown in Figure 3a. The seat share distributions of the top $5\%$ locally fair plans are comparable to the entire ensemble, sometimes achieving lower variance.

**Number of competitive districts.** The Princeton Gerrymandering Project [4] defines a district as *competitive* if the majority color is at most $53.5\%$ of the total votes. Following this definition, in Figure 3b, we compare the percent of competitive districts in a plan. The locally fair plans produce slightly fewer competitive districts: since larger majorities reduce the number of unhappy voters, finding a deviating group becomes harder. However, there exist fair plans that produce the median

---

[8]If more than the $5\%$ of the plans are locally fair, we take an arbitrary subset of the locally fair plans to serve as the top $5\%$.

percentage of competitive districts in the ensemble in all but one state, and they are comparable with or better than the enacted plan in all states. We present results using a different competitiveness metric in Appendix C.

**Minimum compactness.** Another measure of quality of districting plans is compactness – a non-compact district not only makes less geographic sense, but is also more likely to have been gerrymandered to favor one party over another. Two commonly used metrics to evaluate the compactness in redistricting plans are the average and minimum Polsby-Popper scores [34] of the districts in the plan [4]: For a district $D \in \Pi$, the Polsby-Popper score is defined as $4\pi A(D)/P(D)^2$, where $A(D)$ and $P(D)$ are the area and the perimeter of the planar region $D$, respectively; a higher value implies a more compact district. In Figure 3c, we show that the minimum compactness of the locally fair plans remains comparable to that of the entire ensemble. We present results on the average Polsby-Popper score in Appendix C.

Taken together, our results show that local fairness is compatible with fair seat share and compactness, while sacrificing only a small amount on number of competitive districts.

## 4.4 Actual Plans and Additional Results

We also compute the local fairness of plans actually enacted for previous elections. As it is relatively easy to find a locally fair plan in WI, its actual plan is indeed locally fair, while the actual plans for all other states are not locally fair.[9] Using the unf ranks, the enacted plan falls in the $55^{th}$ percentile (fairer than $45\%$ of the plans) in MD, $73^{th}$ percentile in MI, $84^{th}$ percentile in NC, $29^{th}$ percentile in PA, and $22^{nd}$ percentile in TX. While the MI and NC plans are (somewhat surprisingly) above average in local fairness, they have very few (or no) competitive districts. In general, enacted plans that achieve above average local fairness compared to the ensemble (MD, MI, and NC) have fewer competitive districts, and enacted plans with more competitive districts perform below average in local fairness. On the other hand, our results demonstrate that it is possible to find locally fair plans with a comparable (to the ensemble) amount of districts remaining competitive.

In Appendix D, we show that the locally fair notion is also robust to voting patterns in the sense that there is a large overlap between fair plans identified by auditing the ensemble using the voter data from the 2012 and the 2016 presidential elections for the states of NC, PA, and TX.

## 5    Conclusions

In summary, in contrast to extant global notions of fairness in redistricting such as seat distribution or district competitiveness, the notion of local fairness is an explainable notion that mitigates justified complaints of populations in compact geographic regions. Our experiments show that local fairness is not only possible to achieve in practice, but choosing locally fair plans also does not come at the expense of other global fairness properties.

Several open questions arise from our work. In terms of algorithm design, it is an open question of whether there is an approximation algorithm for either the LF AUDITING or LFP GENERATION problem, and whether such an algorithm could take into account compactness. It would also be interesting to extend our methods to capture additional real-world criteria used in redistricting, such as a penalty for splitting up counties, or a requirement for a majority-minority district. In particular, can fair plans be locally modified so that they remain fair and such real-world criteria are satisfied?

Finally, our exploration of robustness of local fairness to voter turnout is preliminary, since it compares the outcomes between two election data in one state. It would be interesting to extend our work to a stochastic setting, where each individual in the population has a "likely voter" score (or probability to vote), and we need high confidence in the non-existence of a $c$-deviating group.

## Acknowledgments and Disclosure of Funding

This work is supported by NSF grants CCF-2113798, IIS-1814493, CCF-2007556, and CCF-2223870.

---

[9]Note that all experiments use the 2016 presidential election data, while the plans in use are mostly drawn in 2011, except the NC one that is drawn in 2019.

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
