# OpenReview forum: "All Politics is Local: Redistricting via Local Fairness"
_NeurIPS.cc/2022/Conference — NeurIPS 2022 Accept_

### Official Review · Reviewer_hdtK · 2022-07-11

**Rating:** 6
**Confidence:** 3
**Soundness:** 4 excellent
**Presentation:** 3 good
**Contribution:** 2 fair

**Summary:**

The paper introduces a method of creating geographical voting districts composed of smaller precincts which are "locally fair". The focus is on two specific problems: determining whether a given redistricting plan is locally fair, and generating a redistricting plan that is locally fair. Both of these problems are shown to be NP-complete and a polynomial time dynamic programming algorithm is described that determines whether a plan is fair. They use this to solve the problem of generating a fair plan by generating many potential plans and ranking them all according to their fairness. Experiments on data from 4 US states show that this method can identify plans with local fairness. The authors also show empirically that they are able to find locally fair plans which meet other fairness and compactness criteria.

**Questions:**

Line 162 says a happy voter is one who agrees with the colour of their district. Definition 2 refers to a red district containing unhappy red voters which would seem to be impossible. Should line 162 say "precinct" rather than "district"? If not, could you attempt to explain this concept more clearly?

Why is WI not discussed in 4.4?

The checklist at the end of your paper says "N/A" in reference to whether you discussed potential negative social impacts of your work. I have trouble understanding this response. Does this mean you cannot imagine any potential negative impacts from a districting process that generate fair and unfair districts?

**Limitations:**

The authors have not done this. While quite a few readers are likely to be familiar with common ideas about the potential negative impacts of this work some discussion about how this specific process could be misused, either intentionally or not, is still rather important.

**Strengths And Weaknesses:**

Generally, the paper is interesting and well written. The problem is well studied; this work provides a strong follow-up to a very related model (ref 6) and addresses some new dimensions of the local fairness issue. The paper is relevant to an ongoing process within the United States which does benefit from new research in this area. I can imagine the included empirical analysis making the work more approachable to non-technical audiences that might bring the results into practice.

Strengths
- the paper is logically structured and the results are discussed in appropriate detail at various locations (e.g. intro does a good job of highlighting the main points, reasonable choices about what is in appendix vs main paper)
- the problem of redistricting remains a real issue in America and a great deal of people may benefit from new developments in the field
- the concept of local fairness seems a reasonable one to include alongside other fairness concepts
- I found the general idea relatively understandable which is important for practitioners

Weaknesses/ways to improve
- I think your definition of local fairness may have a small but significant typo; or I misunderstand. (see Questions, below)
- the "empirical evidence" on lines 199/200 could be more explicitly connected to something in Section 4
Figure 3(a) - unclear what is good here. Is the claim that the minority group should get 0% of seat share rather than an amount proportional to their size?
3(c) - the large difference between NC/WI and TX/PA doesn't seem to be explained (is it to do with urban/rural divide?)
- more discussion on how this could be misused is needed

---

> ### Author Response · Authors · 2022-08-02
> **Response to Reviewer hdtK**
>
> *Weakness 1 and Question 1.* In definition 2, the feasible district $W$ is not a district in the given redistricting plan $\Pi$; instead, it is an alternative district containing voters from multiple districts in $\Pi$, many of which are likely blue. In other words, $W$ is a "red hypothetical district'' that overlaps multiple real districts in $\Pi$, where at least one of them is blue.
>
> *Weakness 2.* In Figure 3(a) and lines 340-343, we deemed a higher (but still less than 50%) seat share for the minority party as better/desirable. This is because all four states we used (NC, TX, PA, and WI) have had close to 50-50 popular vote totals in recent elections, and thus a minority seat share closer to 50% aligns with the proportional share. In general, we do not suggest a higher (or lower) minority seat share to be more desirable. However, we did not make this clear, and we thank the reviewers for pointing this out. Finally, since seat shares are directly tied to the electoral outcome, a lower variance of seat shares in the set of locally fairest plans implies stability in electoral outcome when the redistricting plan is to be selected from such set, and thus is desirable.
>
> *Weakness 3, Question 3, and Limitations.* Please refer to paragraph 6 (Societal Impact) in the [general response](https://openreview.net/forum?id=WSAWRKVjr5K&noteId=2aAmgqDFGj).
>
> *Question 2.* The actual plan for WI is deemed locally fair by the ensemble approach, and it is the only state where this is the case. As most of the plans satisfy local fairness in WI (see Table 1), the local fairness notion has weaker power in separating plans in WI, and thus we do not consider it in the robustness tests in Appendix D. We did measure the actual WI plan with the other metrics (see Figure 3).

---

### Official Review · Reviewer_oFwU · 2022-07-12

**Rating:** 6
**Confidence:** 3
**Soundness:** 4 excellent
**Presentation:** 3 good
**Contribution:** 3 good

**Summary:**

This paper studies local fairness for redistricting plans. The authors give 2 algorithms for auditing plans for local fairness. Both algorithms are approximate and make different types of errors. In experiments in the main text, the authors use the first (simpler) method to audit real and sampled redistricting plans for real data. They show that in many cases, there exist many locally-fair plans for a number of states. They also show that locally-fair plans tend to satisfy global notions of fairness.  Other statistics of the locally fair plans–like partisanship, compactness, and minority seat shares–-are also reported.

**Questions:**

- Footnote 2 was confusing. Is the issue one of double counting? In the construction where we care about unhappy voters as opposed to population is there still double counting?
- Are there times when plan generation is less scalable than the DP? Are there any instances in which you’d recommend usage of the DP?
- Why were the particular values of c in Table 1 chosen? Is 50% a meaningful choice of c? What other values should be considered?
- In the seat shares result, why is a higher percentage of seat shares more desirable than a share that reflects the percentage of the minority? Why is lower variance an important criteria here?
- For the partisanship result: is it a good thing for partisanship of a plan to decrease with respect to real plans? This confuses me since the text seems to say that the fair plans increase partisanship (over the ensemble), which leads to fewer unhappy voters (a good thing).
- Related to the question of partisanship: Are all locally-fair plans for the same state equivalently good/bad? In more detail, is it possible to have two distinct, locally-fair plans that would yield different numbers of red/blue elected officials? If so, how should one choose between two locally fair plans?
- Are plans used in practice locally fair?


**Limitations:**

- No ethical considerations addressed, but the problem area have high potential societal impact.
- One question which is not directly addressed is whether their notion of local fairness should be a high priority consideration in real-world redistricting.
- The algorithms presented are highly specialized. I’m not sure they are applicable for any other problem areas.
- There is no discussion about selection among many locally-fair plans. Such selections could have significant ramifications.


**Strengths And Weaknesses:**

**Originality:**  This work combines and extends a handful of pieces from previous work rather than making a completely original contribution.
- This work borrows a notion of local fairness from recent work [6].  However that work [6] the authors focus on a 1-dimensional problem setting whereas in this work, the authors focus on a 2-dimensional (planar graph) setting.
- The generate-and-test algorithm proposed in this work relies on the ReCom [15] and is otherwise simple. In the proposed algorithm, first generate a number of redistricting plans using ReCom; then, for each district in each plan generated by ReCom, test whether that district is a c-deviating group in each other plan generated by ReCom.
- The dynamic program proposed is somewhat more complex, but seems to be neither practical (i.e., it has a very high runtime) nor useful in that it does not lead to improvements over the simpler heuristic proposed.
- The contributions are clear and as I can tell, relevant work has been cited.

**Clarity:**
- The paper is very well-written. Most of the ideas are explained clearly and I believe that it would be possible to reimplement the experiments form the description in the text (given access to the ReCom algorithm).
- I found the definition of a deviating group to be confusing and required many re-reads. I would suggest including a Figure that shows: a) a small graph or portion of some graph, b) the districts according to $\Pi$ and c) an alternative district–not in $\Pi$--that shows a deviating group. Instead of (or in addition to) a visualization, it would be helpful to include a definition in words as well, for example: “In other words, a plan is unfair if a new feasible district, $W$, can be constructed such that a c-fraction of the voters in $W$ are unhappy under $\Pi$ but would be happy in $W$.
- 3.2.1 is complicated and at the end of the subsection, it seems like the approach is unnecessary. As written in 1.2 (contributions), the DP seems like it will be one of your proposed approaches. I would suggest a different framing where the DP is used to provide further evidence that the more efficient and simpler method doesn’t conclude that certain plans are fair when deviating groups can be found by a more costly and complicated method.
- The word “properties” appears twice in a row in the last sentence of the first paragraph of the conclusion.

**Quality:** this submission is of relatively high quality.
- In the experiments, the authors goal is to determine whether plans that are locally fair are achievable, and whether these plans are compatible with notions of global fairness. By and large, both of these questions are answered affirmatively in a convincing manner.
- At the same time, in my opinion, a discussion of whether local-fairness for redistricting is a good thing (and why) is missing and necessary. While reading, I lean towards thinking that local fairness is good: more people are “happy” (as the paper refers to them) and global fairness is respected. But looking at the results with respect to increased/decreased partisanship and seat shares, I found myself wondering if locally-fair plans are what we want in practice or not. More specific questions related to this thought are in the Questions for the Authors section of this review.
- The theoretical results in this work are proofs of NP-completeness for the problems studied. No approximation results are proved with respect to the proposed algorithms. In my opinion, such proofs would add to the paper but are unnecessary since the application considered is very specific (i.e., redistricting in the United States) and the experimentation is performed with the relevant data.  It’s unclear whether the proposed algorithms could potentially be used in many other contexts. If so, having approximation guarantees becomes more important (so that practitioners would have some idea of how the algorithms are likely to perform).

**Significance:** the results are important, and hold immediate real-world value, however they are limited to a specific problem area.
- This work follows a small, yet well-established area of work on fair redistricting and gerrymandering.  If accepted, this work would likely be built upon.
- Unlike previous work that focuses on global notions of fairness in districting plans, this work focuses on local fairness, which is conceptually different and requires distinct technical tools and algorithms. This paper gives a new method for applying ideas of locally fair partitioning to 2-dimensional (planar graph) redistricting problems.
- Experimental results are new and constitute a new and interesting way at auditing redistricting plans for fairness.

---

> ### Author Response · Authors · 2022-08-02
> **Response to Reviewer oFwU**
>
> *Question 1.* In footnote 2, we compare counting the total number of voters in deviating regions versus counting the total number of unhappy voters in deviating regions, neither double counts voters. However, it is possible for the same set of unhappy voters to form distinct deviating groups by "pulling" in different sets of other voters.
>
> *Question 2 and Clarity Comment 3.* We thank the reviewer for pointing out that the role of the ensemble-based approach (Section 3.1) and the DP (Section 3.2.1) should not be viewed as two parallel/competing approaches. We agree with the suggestion on the framing, that the DP is best suited for verifying (providing more evidence of) the locally fairness of the plans deemed fair by the ensemble approach.
>
> Due to its higher complexity, the DP can only be faster than the ensemble method when the input graph is very small, which is usually not the case in practice. Therefore, in practical usage, it is better to use the ensemble method as the first step (identifying unfair plans) and than use the DP to further investigate the smaller set of remaining "seemingly fair'' plans.
>
> *Question 3.* 50% is the strictest setting of c, as a deviating group only requires a simple majority. It may be the case that we would like to require deviating groups to have a super majority, and c can be increased accordingly.
>
> *Questions 4-6.*
> In Figure 3(a) and lines 340-343, we deemed a higher (but still less than 50%) seat share for the minority party as better/desirable. This is because all four states we used (NC, TX, PA, and WI) have had close to 50-50 popular vote totals in recent elections, and thus a minority seat share closer to 50\% aligns with the proportional share.  In general, we do not suggest a higher (or lower) minority seat share to be more desirable. However, we did not make this clear, and we thank the reviewers for pointing this out. Finally, since seat shares are directly tied to the electoral outcome, a lower variance of seat shares in the set of locally fairest plans implies stability in electoral outcome when the redistricting plan is to be selected from such set, and thus is desirable.
>
> Please also refer to paragraph 3 (partisanship) and 4 (competitiveness) in the [general response](https://openreview.net/forum?id=WSAWRKVjr5K&noteId=2aAmgqDFGj).
>
> *Question 7.*  Please refer to paragraph 2 (Theory vs. practice) in the general response.
>
> *Limitations.* Please refer to paragraph 6 (Societal Impact) and paragraph 1 (LF redistricting) in the general response.

---

### Official Review · Reviewer_mpC7 · 2022-07-12

**Rating:** 4
**Confidence:** 4
**Soundness:** 2 fair
**Presentation:** 3 good
**Contribution:** 2 fair

**Summary:**

This work defines a new notion of “local fairness” for redistricting. The main idea is that a plan is unfair if there are unhappy voters who lose in their current district, but could have been grouped together in an alternative district to elect a candidate from their party.

Drawing locally fair maps and auditing for local fairness are shown to be NP-hard problems. The authors thus offer two heuristics for the auditing problem and evaluate them empirically.

**Questions:**

I have listed many questions and will not be offended if the authors choose to address only those they believe are most important or where I have made some mistake.


1) Regarding the discussion of the dynamic programming method in Section 3.2.2, why does a deviating group need to be compact? First, compactness of deviating groups is not in the definition provided earlier in the paper. Second, I would argue that the deviating group should not need to be compact. Wouldn’t it be easier to gerrymander if non-compact deviating groups are not considered in defining local fairness? Put another way, suppose there exists a plan which is locally fair and satisfies the standard criteria including compactness. Wouldn’t we want to say that plan is more fair than another plan which has a non-compact deviating group?


2) This fairness measure appears vulnerable to the gerrymandering practice of packing and may even encourage packing. Packing a district with mostly voters of the same party is one way to maximize the number of happy voters who cannot be part of a deviating group. This is mentioned briefly in Section 4.3, but can you speak to this more? In particular, I worry that it could be a lopsided fairness metric that favors rural parties packing urban centers. Could this be remedied by letting voters in packed districts be fractionally unhappy (e.g., a voter in a 75% partisan district is 1/3 unhappy)?


3) Does this fairness metric risk invalidating a “fair” plan that voters prefer? There are a few plans that have been held up as fair such as the NC judges map considered in [22] or the 2018 PA map drawn by the PA supreme court? Have you checked whether these or similar maps are locally fair?


4) Lines 127-129 state that locally fair plans “have comparable seat share outcomes” to those with many deviating groups. However, shouldn’t the fair plans have different seat share outcomes from those which represent partisan gerrymanders? Further, you also note that locally fair plans in your ensemble have similar compactness to those with deviating groups, but couldn’t this be an artifact of ReCom producing similarly compact maps as noted in [34]?


5) In the worst case, there are maps that do not have any locally fair plan. How do we compare maps in this case considering that is intractable to compute how “almost fair” a map is and two similarly “almost fair” maps might make different groups unhappy?


6) I’m aware of at least two recent papers which have some similarities to this one, although they are clearly different enough to co-exist. Can the authors can say more to place their work in context with them? I believe the first paper would not have been available to the authors at the time of submission. However, it is especially relevant because the metric is similar and also involves comparing against an ensemble.

Lin, Jerry, Carolyn Chen, Marc Chmielewski, Samia Zaman, and Brandon Fain. "Auditing for Gerrymandering by Identifying Disenfranchised Individuals." In 2022 ACM Conference on Fairness, Accountability, and Transparency

Campisi, Marion, Thomas Ratliff, Stephanie Somersille, and Ellen Veomett. "The Geography and Election Outcome (GEO) Metric: An Introduction." arXiv preprint


7) On lines 338-339, it says, “We compare properties of the top 5% plans in the ranking (which are most locally fair) against the entire ensemble.” How is this 5% chosen for states in which more than 5% of maps have no deviating groups?



Typos:
Line 52: accessing -> assessing
Line 379: “properties properties”

**Limitations:**

I have noted several limitations above which are discussed somewhat to the paper’s credit, but could be discussed more. The only potential negative societal impact is raised in my question about this method being used to discredit “fair” or “good” maps.

**Strengths And Weaknesses:**

Strengths:

The paper addresses and makes progress on a highly relevant and timely problem in society which is still open despite being well-studied.

The biggest contribution/strength is the new idea of how to measure fairness in redistricting. In my view, it’s a good idea, even though there are some flaws/weakness that I believe could make it unpractical in its current form. There is merit in publishing good ideas that may inspire future work as well as informative debate/criticism. So while I have listed many weaknesses and questions below, one could argue that it is worthwhile to publish the paper for its main idea and let the weaknesses be addressed in future work.

I loved the analysis presented in Figure 2 and felt that was an undersold strength of the paper. These results offer useful insights into political geography in general and the urban/rural divide specifically. It could be of independent interest to areas such as the comparison between different electoral systems using districts. To me, this motivates the local fairness notion almost as much as the idea of using it to audit proposed plans.

The paper is written and presented clearly.



Weaknesses:

The main weaknesses were potential flaws in the fairness metric, some questionable claims, lack of theoretical analysis of the auditing methods, and slightly limited experiments. Overall, I feel that the paper has some great ideas, but isn’t ready for publication.


I have several concerns about the fairness metric which are mainly listed in the questions section of this review.


I found the claims and analysis of the two auditing methods to be flawed as noted below. However, I hope the authors can correct me if I’ve misunderstood something. Overall, I would have liked a better theoretical analysis of the two auditing heuristics.

For the ensemble-based auditing method, there are unsubstantiated claims that the method is “likely” to find a deviating group if one exists or that it “provides high confidence”. However, there is no theoretical argument supporting these claims. On the contrary, it seems possible to produce a plan with a deviating group that the ensemble method is unlikely to find especially if non-compact deviating groups are allowed.

For the dynamic programming in trees audit, the analysis is incomplete. There is a description of how to solve the problem optimally in a tree, but no analysis of what solving the problem optimally in many random spanning trees can tell us about a given plan. It is possible that random spanning trees are very likely or very unlikely to contain deviating groups in their subtrees, but this crucial detail appears to be left out. Then, the dynamic programming approach is used to evaluate the ensemble-based approach, but it’s not clear to me that the dynamic programming approach is more accurate.


The experiments were not a major weakness, but adding some subset of the following would strengthen the paper.

More states. In the four states considered, only 2.8% of TX plans were fair at c = 0.5 and that’s using a fairness heuristic with an unknown false positive rate (positive being fair). This raises questions of whether some states do not admit locally fair plans.

More elections. Many of the ensemble papers use data from senate and gubernatorial elections in different years in addition to presidential election data.

A better competitiveness metric. Average partisanship is not a good measure of competitiveness because it doesn’t tell us how many districts in a plan are meaningfully competitive. There are a number of competitiveness measures in the literature. A simple one to test here would be to pick a threshold (e.g. 55% or 53%) and count the number of districts in a plan with partisanship below

A study using small grid graphs for which all possible districts can be enumerated. This could be used to evaluate the auditing heuristics and to gauge the frequency of maps that do not admit any locally fair plan.


This is a minor weakness, but many details are left to the appendix. For example, it would be good to include some sketch or intuition for the NP-completeness claims even just saying which NP-complete problem is used for the reduction. Many aspects of the dynamic programming approach are also in the appendix. While I enjoyed the introduction, I suggest the authors shorten it to make room for more content later in the paper.

---

> ### Author Response · Authors · 2022-08-02
> **Response to Reviewer mpC7**
>
> *Weakness.* We thank the reviewer for pointing out the wordings appear as there are underlying probabilistic claims that can be substantiated in those sentences. We acknowledge that both the ensemble-based and the DP-based auditing approach are heuristics for which we currently could not provide mathematical guarantees (except general properties, e.g., they make one-sided error).
>
> *Question 1.* We agree with the reviewer that ideally the redistricting plan should be compact with no deviating group of any kind (compact or non-compact). However, it is usually too hard to achieve such a strong property. Deviating groups with low compactness have artificial shapes (or even holes, see Figure 5b in supplementary material) and do not serve well as "hypothetical districts'', and requiring no deviating groups may require compromising further on other desirable properties (e.g., competitiveness, compactness).  We believe holding the districts in the plans and the deviating groups to the same standard is a good balance: A deviating group is valid only if it is at least as compact as the least compact districts in the context. We followed this standard in our experiments. For example, in our NC ensemble, the least compact district has a Polsby-Popper score of 0.14 (lines 664-669), and thus we deemed all deviating groups with a score of less than 0.14 as spurious and disregarded them.
>
> *Question 2.* We are aware of that local fairness encourages higher average partisanships, which can be achieved via packing (see paragraph 3-4 in the [general response](https://openreview.net/forum?id=WSAWRKVjr5K&noteId=2aAmgqDFGj)); however, as empirically shown in the experiments, it remains possible to achieve local fairness without extremely increasing the average partisanship score, which can be used as the criteria in choosing among locally fair plans. We agree with the reviewer that a natural extension of the local fairness definition is to allow different magnitudes of unhappiness, and makes an interesting direction of future work.
>
> *Question 3.* Please refer to paragraph 2 (Theory vs. practice) in the [general response](https://openreview.net/forum?id=WSAWRKVjr5K&noteId=2aAmgqDFGj).
>
> *Question 4.* In Figure 3(a) and lines 340-343, we deemed a higher (but still less than 50\%) seat share for the minority party as better/desirable. This is because all four states we used (NC, TX, PA, and WI) have had close to 50-50 popular vote totals in recent elections, and thus a minority seat share closer to 50\% aligns with the proportional share.  In general, we do not suggest a higher (or lower) minority seat share to be more desirable. However, we did not make this clear, and we thank the reviewers for pointing this out. Finally, since seat shares are directly tied to the electoral outcome, a lower variance of seat shares in the set of locally fairest plans implies stability in electoral outcome when the redistricting plan is to be selected from such set, and thus is desirable.
>
> *Question 5.* In our work, we proposed one measure (the unfairness score) to rank plans that are not locally fair. We do acknowledge that, due to the highly non-convex nature of the problem, there may exist different plans that are comparable in the unfairness score (or any other metric defined on local properties) that lead to different global outcomes or have the local unfairness come from different regions. In such cases where local fairness is not effective in separating the plans, we believe it is up to the policy makers to further examine the plans in other existing (global) fairness notions.
>
> *Question 6.* To place our work in context with the FAccT '22 paper, our paper is similar in the usage of an ensemble in an auditing process. However, there is a fundamental difference between how we use the ensemble and their approach: They directly compare the outcome of the plan in question with the outcome of the plans in the ensemble, while our approach uses the ensemble as merely a way to generate candidate districts (deviating groups). Their proposed metric focuses on packing and cracking, and thereby places more emphasis on the gerrymandering strategies (which detects whether the plan is artificially "gerrymandered"), while our metric focuses on individual voter's justified complaints (detecting whether the plan appears as fair to voters, regardless of how it is drawn). The same can be said for the other arXiv paper. Like our work, these recent works contribute to the increasing attention on local issues in redistricting. We will revise our paper by incorporating the more recent works into our literature review.
>
> *Question 7.* If more than the 5% of the plans are locally fair, we take an arbitrary subset of the locally fair plans to serve as the top 5%. Alternatively, we can take the entire subset of locally fair plans and compare it against the ensemble. We did implement this alternative and found no observable difference in the results.

---

> > ### Comment · Reviewer_mpC7 · 2022-08-08
> > **Response to rebuttal**
> >
> > Thanks for your response. I have added some replies to your comments below. In addition, I noticed that your response did not address the issue of using average partisanship as a metric and wanted to give you a chance to respond to that in case you hadn’t seen that comment.
> >
> > Question 1. Thanks for your response and clarification. I think I understand the reasoning now. In this case, either compactness of deviating groups should be part of Definition 2 of c-locally fair plans or added to some new definition in Section 2 that includes this constraint. It should also be explained and motivated in Section 2 as you have in this response.
> >
> > Questions 2 & 3. I understand that fair maps without packing exist. However, it appears that (1) a packed map could also be called fair under this measure and (2) this measure could potentially be used to justify a packed map over a map which is “more fair” by other standards. To me, that is both a limitation and potential negative social impact that should be discussed and explored more in a work proposing a new fairness definition. For example, the court-mandated map adopted by Pennsylvania in 2018 took PA from a 13R-5D state to a 9R-9D state more in line with the voter demographics of the state. It would be easy to check if your ensemble approach could be used to find a plan that is both more c-locally fair and more partisan (favoring either republicans or democrats) than the 2018 map and thus, could be used to undermine such a map.
> >
> > Question 4. Thanks for the clarification. This should be discussed a bit more in the paper and possibly listed as a limitation.
> >
> > Question 5. I think it’s important to note explicitly in the paper that it is theoretically possible in the worst for no fair plans to exist. I don’t think such a statement appears in the current draft. A simple example of this would be a 3x3 grid graph with a checkerboard pattern of party membership. In the real world, my guess is that Massachusetts could become a state with no locally fair map if it were to have even a slight shift toward the republican party. An experiment exploring this or alternately showing that you can find 0.5-locally fair plans for all states would be very interesting.
> >
> > Questions 6 & 7. Thanks for your response.
> >
> > Re: 2. Theory vs. practice. I think the number of states analyzed is sufficient to show the promising potential of the method, but not sufficient to indicate that local fairness is generally achievable in practice as you have claimed. All 4 states analyzed have close to 50/50 partisanship. Including a few more states with different demographics like Maryland would strengthen the claim.

---

> > > ### Author Response · Authors · 2022-08-09
> > > **Response to Reviewer mpC7**
> > >
> > > We thank the reviewer for pointing out we did not respond to the comment on using average partisanship as a metric for competitiveness. We plan to further test against other competitiveness measures in future versions of our paper.
> > >
> > > For potential packing in redistricting, we believe that this could be partially remedied via the suggestion of extending the unhappiness from binary to fractional, since unhappy voters in "lopsided" districts will contribute more unhappiness than those in close districts. Additionally, redistricting relies on practitioners striking a balance among all desired properties. Through this discussion, we see a more general point: the notion of local fairness is best used for filtering/selecting from a pool of "good plans" that satisfy most if not all existing fairness notions, so that the final plan can be more locally stable. Practitioners should not, instead, set a tight local fairness threshold (a large c) so that few locally fair plans exist for a state, and use it to justify this small set of plans ignoring other measures. We plan to address this in future versions of our paper.

---

> > > > ### Comment · Reviewer_mpC7 · 2022-08-09
> > > > **Acknowledging most recent response**
> > > >
> > > > Thanks for the additional response. I agree that counting all fractionally “wasted votes” from each VTD (full votes for people whose party lost in their district and fractional if their party won by more than necessary) would likely address the issue of packing. However, there are two emergent issues with this suggestion which would need to be investigated. (1) It’s very possible that this would make 0.5-locally fair districts much less likely and a larger c would need to be motivated. (2) Using “wasted votes” could have unintended side effects as noted in Bernstein, Mira, and Moon Duchin. "A formula goes to court: Partisan gerrymandering and the efficiency gap." Notices of the AMS 64, no. 9 (2017)
> > > >
> > > > An expanded paper exploring these questions and those raised across all reviews would be a strong submission, especially to venues such as AAMAS and IJCAI which appear to be the most interested in this type of work. Tightening Sections 1 and 2 and moving half of Section 3 to the appendix would probably accommodate the new content.

---

### Official Review · Reviewer_LUyA · 2022-07-12

**Rating:** 4
**Confidence:** 4
**Soundness:** 3 good
**Presentation:** 3 good
**Contribution:** 3 good

**Summary:**

The paper introduces a new measure for fairness in redistricting. Specifically, the paper introduces the notion of local fairness where a redistricting plan is considered to be locally fair if no feasible district exists that can includes at least 1/2 unhappy voters in the given redistricting plan (where unhappy means in the minority party). The problem of auditing and generating locally fair plans is shown to NP-complete and heuristics based on ensemble (sampling) approaches and dynamic programming are shown.

**Questions:**

My main points to authors are above. Please see points under Weaknesses in the above section. In particular, I think points 3 and 4 are the most important.

**Limitations:**

Authors have adequately addressed the limitations and potential negative societal impact.

**Strengths And Weaknesses:**

$\textbf{Strengths}$:

-I think the notion of local fairness is elegant and as the authors point out has the advantage of highlighting the individuals disadvantaged by the drawing of a given plan.



$\textbf{Weaknesses}$:

1-The paper does not include many mathematical guarantees, but this is understandable given that the problem is difficult and generally forbidding to give guarantees to without significant assumptions.

2-This is not very significant but it seems that compactness is de-emphasized as a requirement for a redistricting plan. Although it seems to be universal in the literature (acknowledging that there isn't an agreed upon definition). I would add compactness as a third point along with the connectedness and population balance in lines (149-153).

3-Lines (368-373): the result on NC is a bit odd. One would certainly like local fairness to detected gerrymandering in a highly partisan plan.

4-Following point 3, given a state is it possible that a locally fair redistricting does not even exist? More generally while the notion is interesting, the paper did not elaborate on where we would expect local fairness to help, etc. Perhaps even toy examples could help here.

5-This is a very recent paper so it is acceptable that the authors don't point it out. But it introduces a fairness measure that is similar:
https://dl.acm.org/doi/abs/10.1145/3531146.3533174

$\textbf{Minor Issue}$:

In line (181), I think it is the case that unf(\Pi)=0 if and only \Pi is locally fair which is just more accurate than.

---

> ### Author Response · Authors · 2022-08-02
> **Response to Reviewer LUyA**
>
> *Weakness 2*. Please refer to paragraph 5 (compactness) in the [general response](https://openreview.net/forum?id=WSAWRKVjr5K&noteId=2aAmgqDFGj).
>
> *Weakness 3*. Please refer to paragraph 3 (partisanship) and 4 (competitiveness) in the general response.
>
> *Weakness 4*. Please refer to paragraph 2 (Theory vs. practice) and 1 (LF in redistricting) in the general response.
>
> *Weakness 5*. To place our work in context with the FAccT '22 paper mentioned, our paper is similar in the usage of an ensemble in an auditing process. However, there is a fundamental difference between how we use the ensemble and their approach: They directly compare the outcome of the plan in question with the outcome of the plans in the ensemble, while our approach uses the ensemble as merely a way to generate candidate districts (deviating groups). Their proposed metric focuses on packing and cracking, and thereby places more emphasis on the gerrymandering strategies (which detects whether the plan is artificially 'gerrymandered'), while our metric focuses on individual voter's justified complaints (detecting whether the plan appears as fair to voters, regardless of how it is drawn).  We will revise our paper by incorporating the more recent works into our literature review.

---

### Author Response · Authors · 2022-08-02
**General Comments**

We thank the reviewers for their helpful comments, questions, and the committee for this opportunity to respond. We first address common questions.

### 1. LF in redistricting
We propose to use local fairness as one way to evaluate a redistricting plan. There is no widely agreed upon notion of fairness in redistricting. We are not suggesting local fairness to be the single criteria that can replace all other notions. Instead, we suggest local fairness be used to complement/augment them. Extant fairness notions focus on global measures (such as the distribution of seat outcomes), while local fairness provides a guarantee to the voters at the individual level. Our experiments suggest that it is compatible with extant fairness notions, as the most locally fair plans in the ensemble are comparable in many other metrics to the entire ensemble, while only incurring a small tradeoff with competitiveness (please refer to the discussion on partisanship/competitiveness below). In practice, there could be other considerations when choosing the "best" plan among many locally-fair plans; we leave the question of weighing these considerations to policy makers.

### 2. Theory vs. practice
The results of prior work (e.g., the negative example in the 1D case in [6]) imply that there exist inputs for which no locally fair plans exist. However, our experiments on real datasets indicate that local fairness is generally achievable in practice. In Section 4.4 (lines 366-370) we evaluated the unfair scores of the actual plans in use. The actual plans for NC, TX, and PA were not locally fair, and the plans for TX and PA were more unfair than the average plan in the ensemble. Though the NC plan is less unfair than the average plan in the ensemble, it has a higher average partisanship than every plan in the ensemble (Figure 3b), making it less than ideal. We note that high average partisanship is not equivalent to partisan gerrymandering or a skewed seat share outcome (see the next paragraph on partisanship). Finally, it is relatively easy to satisfy local fairness in WI (see Table 1), and its actual plan is indeed locally fair.

### 3. Partisanship
We realize that the word "partisanship" has caused confusion. We will remedy this in the next version of the paper. Throughout the paper, we measured a redistricting plan by its "average partisanship" over its districts (lines 344-346), where the partisanship of a district is the percentage of its majority voters. Therefore, this measure is not directly tied to the overall seat share outcome of the plan, and plans with high average partisanship values are not necessarily artifacts of partisan gerrymandering. We note that partisan gerrymandered plans (plans that appear as statistical outliers in terms of their seat share outcome) can easily be detected by existing auditing approaches.

### 4. Competitiveness
Districts with lower partisanship (closer to 50%) are considered more competitive. Less competitive districts are usually considered undesirable as they reduce motivation of citizens to vote. Local fairness is indeed easier to achieve with higher partisanship as it leads to fewer unhappy voters. We thus acknowledge that there is a tradeoff between local fairness and competitiveness. However, in our experiments, we showed (lines 344-354) that local fairness can be achieved with a relatively small increase in the average partisanship (small sacrifice of competitiveness), thereby striking a balance in this tradeoff. An interesting question for future research is to achieve local fairness without compromising on competitiveness.

### 5. Compactness
As there is no universally agreed measure (see lines 182-191), we did not tie our problems to a specific compactness requirement, but we agree that the redistricting plans should be reasonably compact. For the plan generation problem, our proposed framework first generates (and selects from) an ensemble of feasible redistricting plans using extant approaches (lines 201-202). As such, given any specific compactness measure, the policy makers can utilize a corresponding approach (that either theoretically or empirically achieves good compactness on that measure) for the ensemble, see lines 214-223. We believe this approach allows more flexibility than explicitly specifying a compactness measure.

### 6. Societal Impact
As a controversial topic itself, there has been a lot of work devoted to discuss potential negative impacts of redistricting (e.g., the book containing [8]). We believe that including local fairness in the criteria for selecting a plan will reduce the abuse of redistricting, though it may compromise on competitiveness of the districts a little. Our view is that redistricting is a reality in the US, and it is thus imperative to develop principled algorithms and metrics for it. We will incorporate more discussion on this and point the readers to relevant references on the potential negative social impacts of redistricting.

---

### Meta-Review · Area_Chair_K5ku · 2022-08-29

**Recommendation:** Accept
**Confidence:** Less certain

**Metareview:**

The reviewers universally agreed that this paper is timely, interesting, and well written. It has two limitations addressing which would make the paper even stronger. First, being upfront about the heuristic (rather than rigorous) nature of some of the statements, as highlighted by the reviewers. Second, is the investigation of other datasets to further strengthen the empirical section. I urge the authors to make these changes for the camera ready.

**Award:**

No

---

### Decision · Program_Chairs · 2022-09-14

Accept